# Control of *Salmonella* and Pathogenic *E. coli* Contamination of Animal Feed Using Alternatives to Formaldehyde-Based Treatments

**DOI:** 10.3390/microorganisms9020263

**Published:** 2021-01-27

**Authors:** Rebecca J. Gosling, Ian Mawhinney, Kurt Richardson, Andrew Wales, Rob Davies

**Affiliations:** 1Animal and Plant Health Agency, Woodham Lane, Addlestone, Surrey KT15 3NB, UK; Rob.Davies@apha.gov.uk; 2Surveillance and Laboratory Services, Animal and Plant Health Agency, Rougham Hill, Bury St Edmunds, Suffolk IP33 2RX, UK; Ian.Mawhinney@apha.gov.uk; 3Anitox, 7 Regent Park, Northants NN8 6GR, UK; KRichardson@anitox.com; 4Department of Pathology and Infectious Diseases, School of Veterinary Medicine, University of Surrey, Veterinary School Main Building, Daphne Jackson Road, Guildford GU2 7AL, UK; a.wales@surrey.ac.uk

**Keywords:** *Salmonella*, *E. coli*, feed, food safety, biocontrol

## Abstract

This study compared a novel non-formaldehyde combination product developed for pathogen control in animal feed Finio (A), with a panel of three commonly used organic acid feed additive products: Fysal (B), SalCURB K2 (C) and Salgard (D). Products were evaluated for their ability to reduce *Salmonella* Typhimurium DT104 and avian pathogenic *Escherichia coli* in poultry feed. A commercial layer-hen mash was treated with each product and then mixed with feed previously contaminated (via inoculated meat and bone meal) with either *Salmonella* or *E. coli*. After 24 h at room temperature, 10 replicate samples were taken from each preparation and plate counts were performed using a selective agar. All concentrations of product A (0.5, 1.0, 1.5, 2.0 and 2.5 kg per metric tonne (MT)) plus the higher concentration of products B and D (6.0 kg MT^−1^) significantly reduced *Salmonella* counts compared with those in the untreated control group (*p* < 0.05). Product C did not significantly reduce levels of *Salmonella* under these conditions. Because of the poor recovery of *E. coli*, statistical comparisons for this organism were limited in scope, but only product A at the highest concentration appeared to have eliminated it.

## 1. Introduction

Raw ingredients for livestock feed production come from a variety of locations [1] and, especially if there has been any exposure to livestock or wildlife faeces, ingredients can act as a source for non-endemic *Salmonella* serovars and other enteric bacteria, including pathogenic *Escherichia coli*. This has been illustrated by field investigations of feed mills, where the ingredient intake pits were found to be the areas most likely to yield *Salmonella*-positive samples [2,3]. More broadly, contamination by *Salmonella* and other undesirable microorganisms can occur at many of the stages of growing, shipping, processing or storage, potentially resulting in contamination of finished feed [4,5].

*Salmonella* is able to persist for many years in dry environments such as those found in feed mills, grain stores and feed bins, and once it becomes resident it can be difficult to eradicate [2]. Resident strains can enter feed processing equipment, including after critical control steps such as heat treatment, and may multiply in situ. This may lead to intermittent or continuous contamination of compound feed during the milling process [2]. *Salmonella* can also survive in the environment on farms, and if wildlife or rodents have access to feeding systems once feed has been delivered, there is the potential for new contamination of feed at this stage [2,6].

In Great Britain (GB) during 2017, compound poultry feeds were found to contain a range of *Salmonella* serovars. The three most common serovars were Ohio, 13,23:i:- and Senftenberg [7]. Serovars Kedougou, Mbandaka and Montevideo were also isolated regularly. Studies have demonstrated links between feed mill or feedstuff contamination and salmonellas of the same serovar in chickens [8,9,10] and humans [11], and these correlations continue to be seen in surveillance monitoring [7].

*Salmonella* Typhimurium (ST) DT104 has previously been responsible for epidemics in food animals and people in Europe, and continues to be reported both in GB and across Europe. In 2013, ST DT104 formed the largest proportion of ST outbreaks in chickens in GB, and variants were isolated from feed samples [12]. A similar increase was observed in 2016, and this suggested a potential emergence of the DT104 subtype in poultry, while it was previously more commonly isolated from ruminants [7,13,14].

In respect of *E. coli* feed contamination, the focus has been on cattle [15], with a particular interest in the potential for dissemination of *E. coli* O157 [16]. However, *E. coli* is also considered to be one of the principal causes of morbidity and mortality in poultry and is associated with heavy economic losses [17]. Avian pathogenic *Escherichia coli* (APEC) with resistance to extended spectrum beta-lactam antimicrobial drugs (ESBL phenotype) has been reported to have zoonotic potential, with some strains being genetically similar to those isolated from human infections and carrying transferrable multi-drug-resistance plasmids [18,19].

Feed manufacturers regularly test raw ingredients on delivery, in addition to sampling finished feed. However, the collection of a truly representative composite sample from large consignments of feed (from which subsamples are tested) is difficult. Furthermore, the standard method of isolation of *Salmonella* from feed (ISO 6579:2017 Annex A) only reports its presence or absence and does not include a quantification step. In order to further manage the risk of livestock and human disease arising from the microbial contamination of animal feed, manufacturers can apply antimicrobial treatments to ingredients and to finished feed. Chemical treatments can reduce existing contamination and may prevent new contamination further along the supply chain.

Historically, formaldehyde-based products were used in the European Union (EU) to counteract feed contamination under Directive 98/8/EC. A change in legislation resulting in Regulation (EU) 528/2012 then required formaldehyde to be approved as a feed additive under Regulation (EC) No. 1831/2003, and in December 2017 this approval was denied by the EU Commission’s Standing Committee on Plants, Animals, Food and Feed (SCoPAFF). Thus, the use of formaldehyde in feed became illegal within the EU, under Regulation (EU) 2018/183. This raised concerns across the feed production industry, as formaldehyde was considered one of the most effective antimicrobial treatments for animal feedstuffs.

With the loss of formaldehyde as an option for antimicrobial feed treatment, the European animal feed industry is turning to other options. One approach is the use of organic acids (OAs), which are already marketed for this purpose. OA-based products have been shown to have anti-*Salmonella* effects (Table 1 and Table 2), which vary between products, between modes of administration and between feed matrices. The modes of action of OAs have been reviewed [20], and their potential roles include preventing [21,22] or removing contamination in feed [23,24,25,26] and directly reducing bacterial load within poultry, the last mainly by activity in the crop [27,28,29,30]. This final effect has prompted anxiety that there might be a suppressive effect of OA-treated feed on orally administered live *Salmonella* vaccines. There appear to be no published studies investigating this, but anecdotal reports relating to non-OA plant antimicrobial additives in poultry feed have not provided evidence of an adverse effect on *Salmonella* control in vaccinated birds [31].

The objective of the present study was to investigate the efficacy of a panel of OA-based products against poultry feed experimentally contaminated with *Salmonella* and *E. coli* by the quantitative measurement of microbial reduction.

## 2. Materials and Methods

Figure 1 provides a diagrammatic guide to the sequence and timing of the experimental process.

### 2.1. Challenge Strains

An isolate of ST DT104 collected from GB broiler chickens in 2014 and a poultry-typical strain of APEC originally isolated from diseased poultry and obtained from the Animal and Plant Health Agency archive, with ESBL phenotype and multi-locus sequence typing profile ST131, were selected. Both were cultured on blood agar plates for 24 ± 3 h at 37 °C. A single colony from each plate was then sub-cultured in 10 mL nutrient broth for 24 ± 3 h at 37 °C. Viable cell density was then determined by spread-plating 1 mL of a serially diluted aliquot onto fresh nutrient broth plates, followed by incubation as above and colony counting.

### 2.2. Feed Inoculation

For both *Salmonella* and *E. coli* experiments, a challenge feed was prepared by adding 10 mL of a 1:9 dilution of the overnight broth culture in phosphate-buffered saline (PBS) to 10 mL of finely ground sterilised meat and bone meal (MBM; 50% crude protein, supplied by the sponsor company). This inoculated MBM (2 g) was then mixed into 198 g of layer crumbles feed (Purina Layena Sunfresh recipe crumbles complete feed for laying chickens, 16% crude protein, Grey Summit, MO, USA), previously ground to 1 mm particles. The resulting contaminated challenge feed samples were stored at 4 °C until used the following day. Bacterial counts were determined by plate culture of serial dilutions of the overnight broth cultures and also of suspensions of the challenge feed on the day of inoculation.

### 2.3. Product Treatment of Feeds

The panel of liquid products used, and their in-use concentrations, are detailed in Table 3. An identification code was given to each product for ease. Concentrations chosen for products B–D were at the request of the sponsor, guided by unpublished in-house studies, and within or above the respective manufacturer’s inclusion rate ranges.

All products were commercially available in Great Britain under the given names at the times of the study and of the manuscript preparation. Each product was diluted to the required concentration on the day of application. As very small amounts of product were being used to treat small batches of food, dose calculations were rounded up to the next even number to allow for equipment limitations. Products were applied to 2.5 kg of un-inoculated layer mash feed, ground to 1 mm particle size (Organic Layer Mash, 16% crude protein, Countrywide, Evesham, UK). Each product was applied using a laboratory-scale feed mixer equipped with an atomizing spray nozzle as a fine aerosol, droplet size of mean diameter 50–60 µm, at eight pounds per square inch (55 kPa) pressure.

### 2.4. Feed Treatment

One day after product application, for each combination of product plus concentration, 10 g of the challenge feed was mixed with 990 g of the treated layer mash feed for five minutes in a laboratory mixer, then held at room temperature (approximately 19 °C) for 24 h. Then, 10 samples of 10 g were taken from each treated batch and each was suspended in 90 mL buffered peptone water (BPW). An aliquot of 100 µL was plated onto xylose lactose Tergitol™ 4 agar (XLT-4; 223410, Difco, Oxford, UK) for *Salmonella*, or ESBL chromogenic agar (CHROMagar ESBL, Paris, France) for *E. coli*. Plates were incubated for 18 h ± 2 h at 37 °C and colony numbers were counted. Up to three replicate plates were made for each 10 g sample, providing a mean count per sample. A negative control of untreated layer mash feed mixed with un-inoculated MBM was included, along with a positive control of untreated but experimentally contaminated feed.

### 2.5. Data and Statistical Analysis

The mean of the log_10_-transformed colony-forming units (CFU) per 10 g sample taken 24 h post-challenge was used as the measure of efficacy. Nil counts were given a value of 0.1 for log transformations. Conventional statistical significance (*p* < 0.05) was applied. Analysis of covariance with product type plus product concentration as a continuous variable was used to assess the effect of the concentration on the log count. Because of the unbalanced number of concentrations between products in the study design, an ANOVA was applied with a Dunnett’s test using 13 independent variables with a 5% level of significance. Contrasts from the ANOVA were also made between product A and each of the other products. Separate comparisons were done at “high” and “low” inclusion rates, these being 3.0 and 6.0 kg per metric tonne (kg MT^−1^), respectively, for products B to D, and 1.0 and 2.0 kg MT^−1^ for product A.

Owing to the low recovery of *E. coli*, and therefore low and skewed CFU counts, a Kruskal–Wallis test for equality of ranks was used to compare the effect of the treatments on *E. coli* based on the counts. In addition, the proportion of positive samples (where recovery of *E. coli* from any one of the replicate plate counts per sample was regarded as a positive sample) was calculated, and a Fisher’s exact test on this statistic was also used to assess the difference between products. For product A, a logistic regression on the proportion positive was used to assess the concentration effect.

## 3. Results

Each experimental inoculating strain produced an overnight broth count of around 1 × 10^9^ CFU mL^−1^. The *Salmonella*-contaminated challenge feed was calculated to contain 1.4 × 10^5^ CFU g^−1^ before being mixed with the product-treated feed. The *E. coli*-contaminated challenge feed was calculated to contain 3 × 10^4^ CFU g^−1^ before being mixed with the treated feed. No *Salmonella* or *E. coli* was recovered from the negative controls, confirming the non-contamination of the feed and the testing process.

### 3.1. Salmonella

For *Salmonella* DT104 the untreated control feed returned the highest mean log_10_ CFU count after 24 h (1.94), and significant reductions (mean log_10_ count less than 1.17) were seen with product A at 0.5, 1.0, 1.5, 2.0 and 2.5 kg MT^−1^, and with products B and D each at 6.0 kg MT^−1^. Significant reductions were not observed with product C, at either 3.0 or 6.0 kg MT^−1^, nor with the lower (3.0 kg MT^−1^) concentration of products B and D. The data for *Salmonella* are presented in Table 4 and illustrated in Figure 2. In direct comparisons between product A and the other products at “high” and “low” concentrations, all other products reduced *Salmonella* counts to a lesser extent than did product A (Table 5). The differences were statistically significant for all except product D at the low concentration.

### 3.2. E. coli

For *E. coli*, recovery was lower than that of *Salmonella* despite similar viable inoculum concentrations, and many samples had nil *E. coli* counts (Figure 3). The mean of the log-transformed counts from positive control (inoculated but untreated) feed was 0.36 after 24 h, but *E. coli* was only recovered from half of these samples, indicating a high die-off of the organism during the test. Owing to this, statistical analysis was limited. A Kruskal–Wallis test based on the mean sample counts from only the untreated control and product-A-treated preparations at each concentration indicated a difference between treatments (*p* = 0.02). However, there was no difference in the *E. coli* counts between “low” (1.0 kg MT^−1^) and “high” (2.0 kg MT^−1^) concentrations of product A.

With product A, a concentration-dependent increase in the proportion of negative samples was observed (Figure 2; *p* = 0.004). There was no significant difference in the proportion negative between products regardless of concentration, despite there appearing to be an increase in negative samples in feed treated both with product A and with the higher (6.0 kg MT^−1^) concentration of product B. There appears to be little evidence of an effect for products C and D.

## 4. Discussion

The microbial doses in the present study were chosen to represent the modest challenge intensity likely to be encountered in feed mills. Furthermore, strains of *Salmonella* (ST DT104) and *E. coli* (APEC) were selected that had contemporary relevance both to GB poultry feed contamination and to public health, including the transmission of antimicrobial resistance. A field strain of ST DT104 that was known to colonise poultry was selected, the birds being an important link in the chain from poultry feed to zoonotic disease. Although this strain was not known to be of feed origin, it survived well when incorporated into feed, and showed sufficient resistance to antimicrobial products to perform the comparative trial of treatments.

The results demonstrate that the treatment of feed within manufacturers’ recommended concentration ranges can produce substantial and significant reductions in *Salmonella* counts over 24 h. However, the findings are consistent with the variability previously observed amongst OA-based preparations (detailed in Table 1 and Table 2), with the trialled products demonstrating differing capabilities to counter *Salmonella* contamination of feed under the present testing regime.

The low recovery of *E. coli* even among positive control samples was likely to have been caused by desiccation as a result of the high level of dust noted in the feed used for the treatment application. The resistance to desiccation of the APEC strain used in the present study is not known. Studies have demonstrated that *E. coli* can survive well in dust samples from poultry houses [32]; however, information on survival in feed is limited to cattle studies [15], where optimum recovery of *E. coli* from dry feed involved resuspension of the contaminated feed in broth overnight rather than resuspending and direct plating as performed in the current study.

Notwithstanding the limitation on assessing effects on *E. coli*, a concentration-related reduction was observed with product A, and the highest concentration of product B also appeared to produce a reduction. The biological relevance of these effects is unclear, given the overall poor recovery of the organism and uncertainty on the role of feed as a risk factor for transmission of *E. coli*, especially of APEC ESBL strains. Further screening of poultry feedstuffs would provide more information on this matter.

For both *Salmonella* and *E. coli*, the antimicrobial effects reported were after 24 h contact time at room temperature. However, a more pronounced effect might have been observed if a longer contact time had also been trialled, as reported by Iba and Berchieri [24]. The degree of efficacy that might be observed more generally with OA-based treatments will depend on the particular circumstances of product and feed composition and probably other factors such as moisture, temperature and natural versus experimental contamination of feed [33].

Awareness of the effect of bacteriostatic agents on the recovery of organisms is important when determining product efficacy [34], and the possibility exists that OA and other product components suppress the detection of viable bacteria by the recovery system (i.e., masking) rather than having truly bactericidal effects. Carrique-Mas et al. [25] reported masking to be greatest when high numbers of *Salmonella* were present in the feed. Masking can be caused by the OA lowering the pH of the culture media, thus causing injury or death to *Salmonella* during culture, with the effect of this varying between serovars and feed matrices [35,36]. Although no neutralisation step specifically to counteract such a phenomenon was included in the current study, at the end of the exposure period the feed samples were suspended in excess BPW (90 mL to 10 g feed) and then 100 μL aliquots were immediately placed on solid media. Thus, any carried-over OA or other components would be at low concentrations and also subject to a further reduction in concentration via diffusion (and, potentially, neutralisation) in the solid media. Therefore, it is considered that any masking effects would be minor, although some differential masking between products cannot be discounted entirely.

There is no universally agreed definition of efficacy in chemical decontamination of feed. Axmann et al. [26] reported that the Austrian Agency for Health and Food Safety considers a decontamination product to be effective if no growth is observed in 10 repeat samples. Product A at 2.5 kg MT^−1^ was the only treatment to satisfy this criterion against the APEC *E. coli*, and was also the treatment closest to this standard (9 out of 10 samples negative) against the *Salmonella* contamination. A formaldehyde-based feed treatment was not included in the test panel, but other studies have reported formaldehyde to be the most effective agent, resulting in a greater than three log_10_ reduction in *Salmonella* counts after 24 h [21,25]. If replicated in the present study, such performance would have resulted in few (if any) positive samples post treatment (Figure 1). In a comparative study including a formaldehyde-based treatment [21], the next most effective product was a medium-chain fatty acid (three log_10_ reduction after 24 h), followed by short-chain OA-based products that achieved a one or two log_10_ reduction at most. Some modest reductions were also observed in the present study. Between-study comparisons are difficult owing to products often only being identified by the active compounds rather than commercial names, but it appears that currently available OA-based products do not replicate the intensity of the antibacterial effect observed in products containing formaldehyde.

## 5. Conclusions

The present study indicates that non-formaldehyde-based decontaminant feed treatments vary in efficacy for reducing the bacterial pathogens *Salmonella* and *E. coli*. The ban on the use of formaldehyde as a feed treatment in the EU may be having an undesirable impact on the presence of *Salmonella* in the livestock feed chain, but close monitoring by the feed industry may provide timely warning of any developing problems in this respect.

## Figures and Tables

**Figure 1 microorganisms-09-00263-f001:**
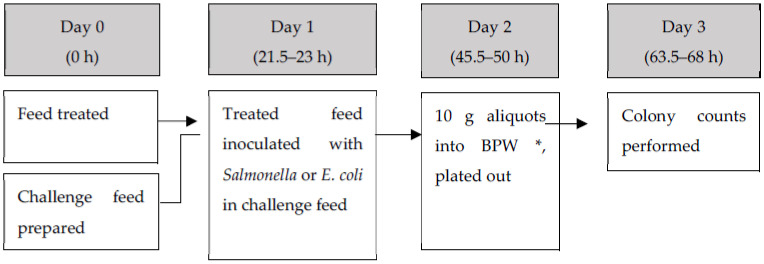
Diagrammatic breakdown of the testing method. * Buffered peptone water.

**Figure 2 microorganisms-09-00263-f002:**
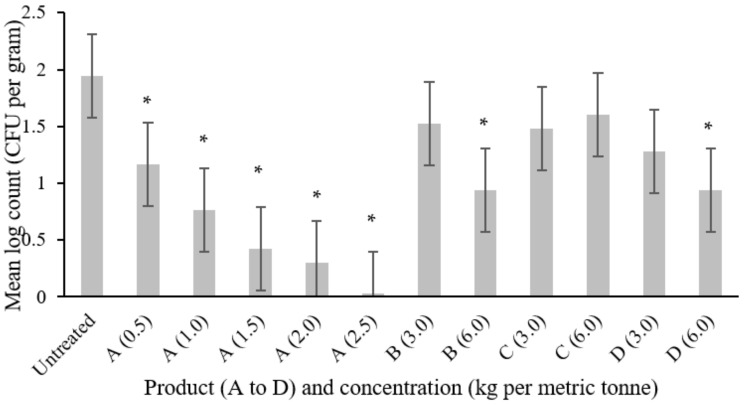
Log_10_ count of *Salmonella* DT104 in feed treated with each product after 24 h contact with products as described in Table 2. * indicates statistically significant difference from the untreated control using Dunnett’s comparison. Error bars indicate 95% confidence intervals.

**Figure 3 microorganisms-09-00263-f003:**
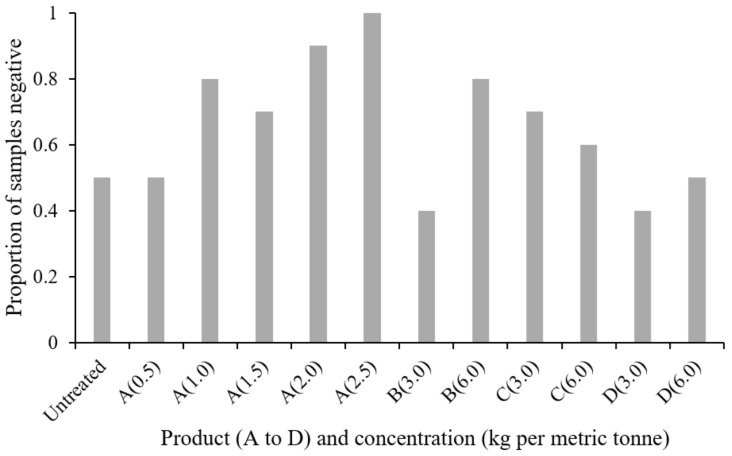
Proportion of samples negative for *E. coli* following 24 h contact with feed treated with each of the products.

**Table 1 microorganisms-09-00263-t001:** Studies on the use of microbicidal additives to reduce *Salmonella* contamination in animal feed.

Matrix	Challenge	Product(s) *, Components, Inclusion Rate(s)	Methodology	Outcome	Ref.
Broiler feed	*Salm*. Kedougou, 10^4^ and 10^5^ CFU mL^−1^	Commercial product: formic and propionic acid (BPO12, BP Chemicals, 0.5% to 0.68%)	Feed inoculated, treated, fed to birds and *Salmonella* monitored in feed up to 3 weeks	BPO12 reduced percentage of positive feed samples but only significantly (compared to control) at 2 weeks	[23]
Broiler breeder feed	Natural contamination	Commercial formic acid product, 0.5%	Feed treated and delivered to (*Salmonella*-positive) farm	Treatment reduced positive feed samples from 4.1% to 1.1%.	[8]
Broiler mash	*Salm*. Typhimurium, 10^9^ CFU mL^−1^	Commercial formic and propionic acid product (Bio-add) at manufacturer’s inclusion rate	Feed inoculated, treated, held at room temperature for 7 days	*Salmonella* counts reduced by 2.5 log_10_ units compared to control.	[24]
Fishmeal/meat and bone meal	*Salmonella* serovars Enteritidis, Typhimurium, Senftenberg and Mbandaka, 10^2^ to 10^4^ CFU 100 g^−1^	Commercial products: formaldehyde (33%), propionic acid and terpenes (1%); formic and propionic acid (1.5%); propionic, formic and sorbic acid in liquid (1.5%) or granule (1.5%) form	Applied 4 h after challenge. Recovery of *Salmonella* at 24 or 72 h post treatment	The 33% formaldehyde product was most effective. Other products had limited effect, especially when a neutraliser was used in recovery	[25]
Range of protein meal types	*Salm*. Typhimurium, starting concentration not stated	Formaldehyde product (0.3%); medium-chain fatty acid product (2%); essential oil blend (2%); lactic, propionic, formic and benzoic acid blend (3%); sodium bisulphate (1%)	Feed inoculated, treated then tested periodically up to day 42 post treatment	The formaldehyde and medium-chain fatty acid products reduced counts immediately across a range of matrices; these remained significantly lower than the declining control counts	[22]
Broiler feed components	Known *Salmonella* contamination	Commercial products, 1% to 7%. Formic and lignin sulfonic acid, liquid (A). Formic and lactic acid, sodium formate, essential oils, liquid (B). Formic, acetic, and propionic acids, ammonium formate, aromatic compounds, liquid (C). Formic and propionic acid, ammonium and sodium formates, liquid (D). Formic, citric, lactic, benzoic and propionic acids, powder (E)	Products added to feed naturally contaminated with *Salmonella*. Contact time of 1, 2 or 7 days	Greater anti-*Salmonella* effect with greater exposure (concentration and contact time), but variation of effect between product and substrate. *Salmonella* not detected after 6% of product B. Products A, C and D most effective in in corn gluten. Product E generally ineffective.	[26]

* Commercial names are provided where documented in literature.

**Table 2 microorganisms-09-00263-t002:** Studies on the use of microbicidal additives to prevent *Salmonella* contamination of feed.

Matrix	Challenge	Product(s) *, Components, Inclusion Rate(s)	Methodology	Outcome	Ref.
Broilermash	*Salm*. Typhimurium, 10^9^ CFU mL^−^^1^	Commercial formic and propionic acid product (Bio-add) at manufacturer’s inclusion rate	Feed treated then challenged at time points up to 28 days	Reduced counts by 2.5 log_10_ units compared to control with challenge up to 28 days	[24]
Home-ground mixed grain	*Salm*. Enteritidis, 10^7^ CFU mL^−^^1^	Eleven unnamed products (E to M) tested. Only four achieved ≥2 log reduction: multipurpose feed acidifier (E, 0.45%), medium-chain fatty acid blend (F, 0.3%), detergent, organic acid and salts blend (H, 0.2%); formaldehyde, propionic acid, terpenes and surfactant blend (M, 0.3%)	Feed treated then challenged. Recovery of *Salmonella* at 24 h and 7 days post challenge	>3 log_10_ (M), 3 log_10_ (F) and 2 log_10_ (E) reductions after 24 h. Two log_10_ reduction seen after 7 days with H	[21]

* Commercial names are provided where documented in literature.

**Table 3 microorganisms-09-00263-t003:** Organic acid products and treatment concentrations.

Product Details	Identification code	Formulation	Inclusion Rate (kg tonne^−1^)	
			Advised *	Present Study
Finio (Anitox, Lawrenceville, GA, USA)	A	Phytochemicals and carboxylic acids	0.5 to 2	0.5 (0.05%)
				1.0 (0.1%)
				1.5 (0.15%)
				2.0 (0.2%)
				2.5 (0.25%)
Fysal (Selko, Tilberg, The Netherlands)	B	Blend of organic acids with their ammonium salts	1 to 3	3.0 (0.3%)
				6.0 (0.6%)
SalCURB K2 (Kemin, Herentals, Belgium)	C	Blend of formic, lactic and propionic acid, salts and a surfactant	3 to 6	3.0 (0.3%)
				6.0 (0.6%)
Salgard SW (Anpario, Worksop, U.K.)	D	Blend of propionic acid and ammonium salts of propionic and formic acids	1 to 8	3.0 (0.3%)
				6.0 (0.6%)

* Manufacturer’s recommendation. Typically, inclusion rates are selected according to feed type, livestock species and other risk factors. The range given is for compounded feeds.

**Table 4 microorganisms-09-00263-t004:** *Salmonella* counts in product-treated feed 24 h after microbial challenge.

Product (See Table 3)	Inclusion Rate (kg MT^−^^1^)	Viable Counts (CFU g^−^^1^)		Significant Under Dunnett’s Test *
		Mean of Log_10_ Values	Std. Deviation of Log_10_ Values	
A	0.5	1.16	0.601	Yes
	1.0	0.76	0.500	Yes
	1.5	0.42	0.495	Yes
	2.0	0.30	0.284	Yes
	2.5	0.03	0.091	Yes
B	3.0	1.52	0.995	No
	6.0	0.94	0.594	Yes
C	3.0	1.48	0.826	No
	6.0	1.61	0.662	No
D	3.0	1.28	0.416	No
	6.0	0.94	0.425	Yes
Controls:				
Negative	-	0.00	0.000	n/a
Positive	-	1.94	0.533	Ref.

* *p* ≤ 0.05, test of difference from *Salmonella* counts in positive control preparation. n/a not applicable. Ref. baseline reference value for comparisons.

**Table 5 microorganisms-09-00263-t005:** ANOVA predicted difference (contrast) in mean log_10_
*Salmonella* counts for all products at low and high concentrations against product A.

Product Comparison	Contrast	Std. Error	z	*p* > |z|	95% Confidence Interval
Low concentration *					
B vs. A	0.718	0.234	3.07	0.002	0.259 to 1.177
D vs. A	0.447	0.234	1.91	0.056	−0.012 to 0.906
C vs. A	0.605	0.234	2.58	0.010	0.146 to 1.064
High concentration ^†^					
B vs. A	0.683	0.179	3.82	0.000	0.332 to 1.035
D vs. A	0.633	0.179	3.54	0.000	0.282 to 0.985
C vs. A	1.210	0.179	6.75	0.000	0.859 to 1.561

* Product A at 0.1% *w*/*w*; products B, C and D at 0.3% *w*/*w*. ^†^ Product A at 0.2% *w*/*w*; products B, C and D at 0.6% *w*/*w*.

## Data Availability

Data is contained within the article.

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
