# Peer review of "Control of Salmonella and Pathogenic E. coli Contamination of Animal Feed Using Alternatives to Formaldehyde-Based Treatments"

_microorganisms, 2021, doi:10.3390/microorganisms9020263_

Round 1

Reviewer 1 Report

The paper illustrates an extremely interesting and current topic for the safety of farm industry and production. The tests described were conducted with rigor and technical precision, supported by numerous statistical evaluations suitable for the purpose of the experiment.

however I would need more explanations on the choice of treatments to be tested, in order to make these decisions more understandable:

• as briefly reported in table 1, why the various compounds are tested at different concentrations (5 different levels, from 0.05% to 0.25% for compound A, only two, 0.3% and 0.6%, for the others), regardless of the manufacturer's instructions ?

• the legend of table 1 explains that "inclusion rates" are chosen "on the basis of the type of food, species and other risk factors", but does not describe any. please, is it possible to have more clarity?

• the formatting of table 1 itself makes it difficult to read and interpret without using the text (part of the text in bold without explanations, difficulty in following the alignment of the lines): please, could you amend it?

• furthermore, regardless of the instructions of the manufacturers on the use of the compounds under test, if the aim is to understand which compound can be more efficient, I think a statistical comparison between different concentration levels could be quite strange (tab 4): please, could you give some explanation?

fig. 3 would benefit from a greater distance between the title and the labels shown on the abscissa axis

the bibliography section is adequate and correctly reported

Author Response

Thank you for the helpful comments. These have been addressed as described in the attached document.

Reviewer 2 Report

Manuscript No. microorganisms-1084719: “Control of Salmonella and Pathogenic E. coli Contamination of Animal Feed Using Alternatives to Formaldehyde-Based Treatments”

This is a very interesting study on the efficacy of a panel of organic acid based products against poultry feed experimentally contaminated with Salmonella and Escherichia coli, by quantitative measurement of microbial reduction.

The study is well organized and the obtained results well discussed; The conclusions aresupported by the results.

Comments to the Authors:

It is well done work, thank you.

Author Response

No changes required. Thank you for your comments.